# Review on Modification of Glucomannan as an Excipient in Solid Dosage Forms

**DOI:** 10.3390/polym14132550

**Published:** 2022-06-23

**Authors:** Nuur Aanisah, Yoga W. Wardhana, Anis Y. Chaerunisaa, Arif Budiman

**Affiliations:** 1Department of Pharmaceutics and Pharmaceutical Technology, Faculty of Pharmacy, Universitas Padjadjaran, Jatinangor 45363, Indonesia; nuur20002@mail.unpad.ac.id (N.A.); anis.yohana.chaerunisaa@unpad.ac.id (A.Y.C.); arif.budiman@unpad.ac.id (A.B.); 2Department of Pharmacy, Faculty of Mathematics and Natural Sciences, Tadulako University, Palu 94118, Indonesia; 3Study Center Development of Pharmaceutical Preparations, Faculty of Pharmacy, Universitas Padjadjaran, Jatinangor 45363, Indonesia

**Keywords:** glucomannan, chemical modification, physical modification, excipient

## Abstract

Glucomannan (GM)—a polysaccharide generally extracted from the tuber of *Amorphophallus konjac*—has great potential as a filler–binder in direct compression, disintegrant in tablets, or gelling agent due to its strong hydrophilicity and extremely high viscosity. However, it has poor water resistance and low mechanical strength when used as an excipient in solid form. Several physical and chemical modifications have been carried out to improve these drawbacks. Chemical modification affects the characteristics of GM based on the DS. Carboxymethylation improves GM functionality by modifying its solubility and viscosity, which in turn allows it to bind water more efficiently and thus improve its elongation and gel homogeneity. Meanwhile, physical modification enhances functionality through combination with other excipients to improve mechanical properties and modify swelling ability and drug release from the matrix. This review discusses extraction of GM and its modification to enhance its applicability as an excipient in solid form. Modified GM is a novel excipient applicable in the pharmaceutical industry for direct compression, as a tablet disintegrant, a film-forming agent, and for encapsulation of macromolecular compounds or drug carriers for controlled release.

## 1. Introduction

Solid dosage of drugs is most preferable because it provides accurate dosage and is more stable than other forms [1]. Common uses include uncoated and film-coated tablets and film. Production requires polymers to enable pharmaceutical products to optimally control drug release [2,3] and to improve physicochemical properties [4,5]. Natural polymers such as glucomannan (GM) have attracted extensive attention due to their biodegradability, nontoxicity, harmlessness, and biocompatibility.

Glucomannan (GM) is a polysaccharide typically extracted from *Amorphophallus oncophyllus* [6] and *Amorphophallus muerelli* Blume [7]. It has the ability to thicken and form a gel; hence, this compound is widely used in various industries, including the pharmaceutical industry as a binder [8], thickener [9], gelling agent [10], film former [11], coating material for tablets [12,13], emulsifier [14], and stabilizer [15]. 

As a natural polymer, GM has properties that are superior to other polysaccharides when used as excipients for solid preparations, especially in tablet production. GM could be the excipient of choice for direct compression—the most efficient tablet manufacturing method—because it has desirable free-flowing and compressibility behavior [16,17,18]. GM is also reported as a widely used coating material and stabilizer in the pharmaceutical industry due to its gelling properties and particular rheological properties [11,13,19].

Native GM has several disadvantages for pharmaceutical applications, such as extremely high viscosity and low mechanical strength [20,21]. In addition, GM’s high-water absorption index causes poor water resistance and limits some potential applications [14,22]. However, these shortcomings of native GM could be overcome through chemical or physical modification to enhance its structural and functional quality. 

Chemical modification involves the substitution of functional groups in GM’s structure including esterification and etherification and elongation of the molecular chain through the formation of crosslinks and encapsulation. Depending on the degree of substitution (DS), these modifications alter several characteristic of GM, such as homogeneous film formation [11], increased tensile strength [15], improved thermal stability [15], and sustained release [23]. 

GM can be physically modified to improve functionality without undergoing chemical changes. Physical modifications involve mixing native GM with other excipients through processes involving milling [24], moisture [25], temperature [26], pressure [27], radiation [28,29], etc. Physical modifications cause variations to particle size, shape, surface properties, porosity, density, and to functional properties such as swelling capacity and gelation ability. These modifications directly influence disintegration and mechanical properties when used as an excipient in solid form.

Therefore, the objective of this review is to discuss several extraction methods of plant-derived GMs from different sources and the effects of chemical and physical modification of GM on its physicochemical characteristics in order to explore its potential, especially as a raw material for solid dosage excipients, which are widely needed by the pharmaceutical industry.

## 2. Methodology

The review was based on studies identified in the Scopus and PubMed electronic databases using the keywords “glucomannan”, “modification”, “extraction”, and “application” in the time range of 2011 to 2021. The specific keywords related to application of glucomannan as an excipient in solid form were “chemical modification”, “physical modification”, “tablet”, and “films”. Meanwhile, reviews, studies written in non-English languages, original articles published before 2011, studies on unrelated topics such as glucomannan activity on health, and clinical studies were excluded. Abstracts were read and excluded if the reported article did not mention any possible application as excipients in solid form. Finally, 85 articles were fully read and included in the present paper. A flowchart of article collection is shown in Figure 1.

## 3. Polysaccharides from Different Sources

### 3.1. Natural Polysaccharides

Natural polysaccharides can be derived from four main sources: algae, plants, animals, and microorganisms (Figure 2). Demand for polysaccharides from natural origins is tremendously because they provide several benefits, including natural abundance, ease of isolation, and ability to be chemically modified in order to fit technological demand. Furthermore, these polymers may be enzymatically hydrolyzed, and they produce a noncarcinogenic, degradable product [30,31]. 

Natural polysaccharides consist of many monosaccharide residues joined by O-glycosidic bonds. When hydrolyzed, polysaccharides produce simple sugars such as glucose, galactose, mannose, arabinose, xylose, uronic acid, etc. Among numerous sources of naturally occurring substances, plants are considered a potentially renewable resource since they can provide a certain amount of natural polymers of plant origin [32,33]. Figure 3 shows some essential steps in the isolation and purification of polysaccharides from plants.

### 3.2. Petrochemical Synthesis

Structurally, polysaccharides are composed of several monosaccharides joined together by O-glycosidic linkages. O-glycosidic linkages are formed by dehydration of the hemiacetal hydroxyl group of one sugar (a glycosyl donor) with a hydroxyl group on the anomeric carbon of another sugar (a glycosyl acceptor). Due to the presence of multiple hydroxyl groups, one glycosyl acceptor residue can be connected to more than one glycosyl donor via different O-glycosidic linkages. Consequently, polysaccharides may be linear or branched, and branching may occur at different positions of sugar units in the polysaccharide backbone with different branching densities [34,35]. Generally, three methods are used to synthesize polysaccharides: (1) stepwise glycosylation [36,37]; (2) condensation polymerization [38]; and (3) ring opening polymerization [39,40]. 

## 4. Structure and Physicochemical Properties of GM

GM is a natural heteropolysaccharide with a linear chain consisting of D-glucose and/or D-mannose in various proportions linked by β-1,4 glycosidic bonds. It also has multiple branching at β-1,3 glycosidic bonds to mannose units as shown in Figure 4 [41].

The molecular weight varies from 200,000 to 2,000,000 Daltons, giving it incredibly higher viscosity than any known dietary fiber such as guar or locust bean gum [42,43]. When GM sol concentration is below 0.55%, it is only slightly affected by shear rate, indicating Newtonian fluid flow characteristics. However, at higher concentrations, shear rate can affect viscosity, leading to shear thinning and indicating non-Newtonian pseudoplasticity [44]. Based on previous reports, the viscosity of konjac glucomannan solution (1.0 g/100 g) can reach ~30,000 cps [45]. 

GM is a hydrophilic polymer due to the abundance of hydroxyl and carbonyl groups in its molecular chain. The hydrogen bonds between each molecule affect its solubility; hence, the stronger the bonds, the lower the solubility in water. In contrast, low acetyl group branching (5–10% at the C-6 position, i.e., one branch per approximately 19 sugar residues) reduces hydrogen bonding, thereby increasing solubility; this causes high water absorption of 105.4 g/g (water/GM) [46]. Water absorption is also affected by granule size and surface morphology—a reduction in particle size will increase surface wrinkle density, which culminates in higher hydration rates [47].

The formation of gel is by hydration of water; this can be accelerated by heating and vigorous stirring. GM also forms synergistic gels in a thermally reversible reaction with other polysaccharides, such as xanthan gum [48], κ-carrageenan [49], and gum tragacanth [50], which increase the mechanical strength and decrease syneresis. This is presumably due to agglomeration or physical entanglement and dynamic hydrogen bonds with other polysaccharides [48,49,50].

In recent years, GM has attracted special attention from researchers and the food industry due to its bioactive, biodegradable, and hydrophilicity properties. This high-molecular-weight polymer is known as a hydrocolloid and interacts strongly with water [51]. Hydrocolloids are used in the food industry because of their thickening, gelling, stabilizing, texture-modifying, and film-forming properties.

## 5. Extraction Optimization

For extraction, the organic solvents usually used to obtain GM from flour are hot water [52], ethanol [7], or isopropanol sol [53]. As the majority of the glucomannan is present in the cell walls (intracellular polysaccharides) of higher plants, extraction begins with crushing plants to release intracellular polysaccharides. Furthermore, plant cell walls are mainly enclosed by lipids, which are removed by organic solvents, allowing extraction of glucomannan. An illustration of GM’s separation from impurities is depicted in Figure 5. The flour swells in organic solvents, and the aqueous part dissolves impurities trapped in the konjac particle, such as soluble sugars and partial proteins. During heating, gelatinization occurs, which irreversibly dissolves the starch in water, while ash is simultaneously removed with increasing temperature. Therefore, heat affects GM purity, while purified products are precipitated in the presence of an antisolvent [54].

Among other techniques, extraction by ethanol precipitation is the most effective method to obtain GM; this is because its huge particles are difficult to destroy, and the incredibly tough shape is soluble in water but not in ethanol. Based on these characteristics, Yanuriati et al. found an easier and faster method to isolate GM from *Amorphophallus* by repeated milling of the fresh tuber slices using ethanol, followed by filtration without further purification. The results showed exceptionally high purity (90.98%), viscosity (27.940 cps), and transparency (57.74%), and significantly reduced ash and protein concentrations without starch content [7]. Furthermore, several studies used ethanol–water solutions in different concentration gradients to control the solubility of GM [55,56]. Isopropanol works excellently to remove impurities, including starch and carotene [53], while ethanol is also effective for obtaining GM but does not dissolve carotene thoroughly. 

Previous statistical studies applying response surface methodology (RSM) proposed that precipitation efficiency is correlated to several factors, namely, harvest time [7], processing temperature [53,54], and solvent concentration [53,57]. First, the best harvest time to get GM from plants such as *Amorphophallus muelleri* tubers is during dormancy. Plant dormancy is a period of arrested growth. Because GM is one of the energy sources for leaf growth, during dormancy it is not used for metabolic processes and accumulates in the tubers [58]. Second, temperature correlates positively with GM content according to Xu et al., who reported an optimum temperature of 68 °C using 40% ethanol increased purity from 74.13% to 90.63% [53], while temperatures > 78 °C are not recommended because they are higher than GM’s exothermic transition temperature and disrupt the molecular chain [54]. Third, the optimum concentration of ethanol solvent is 50%; this is because it is difficult to remove water-soluble impurities from the flour with concentrated ethanol. Meanwhile, in diluted ethanol, more water is absorbed, which leads to greater swelling, making it difficult to obtain the GM [53].

Several other sources of GM apart from *Amorphophallus* are shown in Table 1 below.

## 6. Chemical Modification

Native GM forms very high viscosity solutions, where the intrinsic viscosity of 1% can reach 30,000 cps, and so it has potential as a good film-forming agent [68]. However, a very viscous external gel layer on the surface of particles immediately after dispersion prevents water penetration and drug dissolution, and thus its application is limited as a carrier for immediate drug release [69]. As a film, it has poor water resistance due to the large number of free hydroxyl and carboxyl groups distributed along the backbone, and it exhibits high moisture absorption. As a result, native GM has the weaknesses of poor water resistance and low mechanical strength [20,21]. 

Several structural modifications of GM have been performed to enhance its structural and functional qualities, including oxidation [70,71,72] and etherification by addition of acetyl [41,49,59,73] and carboxymethyl [2,5,13,15,74,75,76,77,78,79] moieties on hydroxyl groups of GM. Chemical modification with different degrees of substitution (DS) give different physical and mechanical properties. DS is largely affected by the amount of sodium hydroxide and by different dispersion media. Higher degrees of substitution contribute to lower viscosity and particle size, denser network structure, and better tablet strength [80].

GM is an ideal candidate for appropriate modification by chemical functionalization. Each of the glucose–mannose units have reactive hydroxyl groups, which are the major sites for chemical modification. In addition, several studies discovered that chemically modified GMs can be used for the sustained release of drugs [2,23,43]. Among various modification methods such as acetylation [59,73,81], carboxymethylation and oxidation [70,71,82,83], carboxymethylation is the most common and suitable for solid and film dosage [2,3,5,13,74]. The effects of carboxymethylation on GM (CMGM) are described as follows:

### 6.1. Increased Solubility

Chemical modification of CMGM affects solubility; carboxymethylation with NaOH catalyst substitutes chloroacetic acid with a hydroxyl group, which partially replaces hydroxyl and acetyl groups with carboxymethyl [2,74,75,84]. Incorporation of a carboxymethyl group appears as an extending chain structure that reduces hydrogen bonding between the polymer chains and increases the water-binding capacity, as shown in Figure 6 below.

Modification to CMGM improves solubility because excess substitution by carboxymethyl groups breaks extensive hydrogen bonds, leading to a drastic decrease in crystallinity and an increase in solubility [85]. Modification also changes the amount of acetyl located randomly at the C-6 position of the sugar unit. The increase in solubility is due to the incorporation of water-soluble carboxylate groups during deacetylation (Figure 4). Additionally, changes in water-binding properties are caused by reduction and/or loss of crystal structure in the granules, making them mostly amorphous and more hygroscopic [76].

However, based on several studies, the smaller the degree of substitution, the lower the hydrophilicity due to the increase in the contact angle from θ = 48.1° [14] to θ = 97.3 ± 4.2° [80]. Water solubility is also lower in CMGM than in GM. This is because the particles in both are all amorphous; hence, carboxymethylation does not reduce and/or eliminate the crystalline structure inside GM granules, but, rather, alters its granular surface structure, which might affect moisture adsorption [76]. 

Ohya et al. reported a dicarboxy–glucomannan derivative capable of increasing the solubility of GM in water and interacting with other positively-charged polymers [86].

### 6.2. Reduced Viscosity

The concentration and type of polymer in coating solutions affects viscosity [87,88]. High viscosity sols produce nonuniform films due to low diffusivity, giving a “solid skin” that retards solvent evaporation and causes hydrodynamic instability. The solid skin is under mechanical tension and might break, thereby causing variations in film thickness [89]. Consequently, moderate viscosity is desirable for film formation. For utilization as a coating material, several studies suggest viscosity lower than 700 cps [90]. The viscosity of the coating solution can be increased by using polymers with high molecular weight (Mw), such as GM, which averages 200,000 to 2,000,000 Daltons [91,92], giving it the highest intrinsic viscosity compared to other polysaccharides at approximately 30,000 cps at a concentration of 1% [6].

Several methods have obtained GM with low Mw, for example, depolymerization, such as deacetylation, and carboxymethylation with strong bases to break the glycosidic bonds [93]. The viscosity of GM at 25 °C decreased significantly from 4660 cps to <500 cps after modification [2,76].

Deacetylation through carboxymethylation changes the structure from semi-flexible straight chains to elastic microspheres that decrease inherent viscosity (Figure 7) [80]. As a comparison, substitution of carboxymethylation groups in cellulose also affects viscosity. At a concentration of 1%, cellulose and carboxymethyl cellulose (CMC) have viscosities of 240 cps and <100 cps, respectively.

### 6.3. Increased Tensile Strength 

Generally, the presence of more -COO- groups due to carboxymethylation of the CMGM backbone improves gel strength by forming more crosslinks, while a high DS also increases mechanical strength [15]. The introduced COO− group can efficiently bind more water, which can act as a plasticizer to improve elongation of the film [94]. As the DS of CMGM increases, formed pore size decreases and the tissue structure becomes denser, indicating stronger interaction at higher DS [81]. This high density also increases tablet strength [3,24], but an excessive amount of CMGM causes charge repulsion, thereby weakening its mechanical properties [74].

### 6.4. Improved Thermal Stability

CMGM maintains the gel network through hydrogen bonding upon heating to 95 °C for 2 h, implying excellent thermal stability [15]. Carboxymethylation increases the thermal stability of GM in a DS-dependent manner. Based on thermogravimetric analysis (TGA), GM is degraded in three stages. TGA recorded a change in mass due to moisture removal from 60–200 °C. Meanwhile, from 200–300 °C, great weight loss was recorded in GM, CMGM (DS 0.28), and (DS 0.7), with values of 64.16%, 49.73%, and 43.17%, respectively. In the final stage of decomposition at a temperature of 300–500 °C, there was a greater decrease in mass change in GM than in CMGM due to thermal degradation [2].

## 7. Physical Modification

Co-processing is a technique for mixing two or more excipients at the sub-particle level to synergistically enhance functionality and mask undesired properties without undergoing chemical changes [95]. This method can change fundamental characteristics such as particle size and shape, morphology, porosity, density, and surface area, which affects flowability, compressibility, compactibility, and ultimately influences the disintegration and mechanical properties of tablets [96]. Some examples of GM processed together with other excipients are shown in Table 2 for various applications, especially controlling drug release through crosslinking and/or formation of dense hydrogen bonds.

## 8. GM application as an Excipient for Solid Dosage Forms

### 8.1. Direct Compression Excipient

Direct compression is the most preferred tablet manufacturing method because it is effective and efficient for industrial use. The excipient used in this process plays an important role, especially in the formulation of low-dose, active pharmaceutical ingredients (APIs), and is affected by the overall properties of the mixture. Excipients must have good flowability, compressibility, and compactibility.

GM is a candidate for a direct compression excipient because it has good flowability, as demonstrated by an angle of repose of <35°, and the particle size of CMGM (130 µm) [24,99]. Moreover, co-processing with other excipients such as HPMC [97], lactose, and starch [84] creates products that meet these criteria. This is because co-processing combines excipients, thereby increasing particle size, which also affects compressibility. Smaller particles tend to have larger air gaps, which cannot be compressed. Co-processed excipients with larger particle sizes provide more-compact structures because less air is captured in the tablet, reducing the release of elastic energy [5,76].

Based on X-ray results, deacetylated CMGM has an orthorhombic unit cell pattern with spatial plane a = 9.01 Å, b = 16.73 Å, and c (fiber axis) = 10.40, and a possible space group of I222. Given that orthorhombic crystals are not capped at higher compression pressures, this increased compressibility is due to high-level densification or decreased relaxation stress [10,100].

### 8.2. Tablet Disintegrants

A disintegrant is a substance or mixture of substances added to a drug formulation that helps disintegrate tablets into smaller particles, making them dissolve more quickly. Various natural polysaccharides, such as modified starch, agar gum, and guar gum, have been used as disintegrants [101].

Carboxymethylation of polysaccharides increases water absorption and causes faster disintegration. As a comparison, sodium carboxymethyl cellulose (Na CMC) with a DS of 1.0 tends to make CMC easily disaggregated in water, making it more hydrophilic compared to water-insoluble cellulose [102,103]. A similar property was also observed in carboxymethylation of xanthan, where an increase in the degree of substitution exhibits greater hydrophilicity and lower molecular weight [104].

GM is a hydrophilic natural polymer, but its solubility in water can be reduced by the formation of strong hydrogen bonds during purification and drying. Several chemical modifications, such as CMGM, have been carried out to obtain derivatives with better solubility. The increase in solubility occurs due to the incorporation of water-soluble carboxylic groups. Changes to water-binding properties are caused by reduction and/or loss of crystal structure in the granules, making them mostly amorphous and more hygroscopic [76]. Furthermore, water absorption is influenced by DS, where high (>0.5) carboxymethylation increases absorption [9]. CMGM with high-water absorption capacity causes rapid disintegration; hence, it has great potential as a tablet disintegrant. In contrast, low DS values (ranging from 0.2–0.4) obtained with sodium acetate catalyst reportedly reduced water absorption [76].

Ma et al. reported super-disintegration and rapid drug release by the combination of GM–superporous hydrogel composite (SPHC) on an artificial gastric fluid medium, where drug release reached >90%, with GM accounting for 80% in the first 3 min. These results support the potential application of modified GM as a disintegrant for tablet dosage forms containing drugs needing rapid disintegration [105].

### 8.3. Film-Forming Agent

One property of GM solutions is their extremely high viscosity, which needs to be modified either chemically or physically. With lower viscosity, this material is easier to apply as a film.

Xie et al. revealed that the combination of CMGM and crosslinked chitosan can be used in wound dressing due to good swelling ability and moisture permeability, which effectively protect the wound from excessive dehydration and accumulation of exudate. Moreover, the produced film had excellent thermal stability and biocompatibility, which can accelerate tissue regeneration [77].

Apart from wound dressing, the combination of CMGM with other polymers, such as gelatin, is used as packaging films with increased thermal stability. Single application of CMGM showed continuous thermal degradation, whereas the onset temperature increased when it was combined with gelatin. Cross-linkage with gelatin through free amino and carboxyl groups also exhibited electrostatic interactions that improved the mechanical properties of the nanocomposite films [106].

### 8.4. Sustained Release Agent

The advantages of sustained-release tablets include increased plasma drug level stability and patient compliance, leading to optimum therapy [3,23,81,84,99]. One method to control drug release is through a polyelectrolyte complex reaction, which improves and increases gel strength. Mixture of two oppositely charged polysaccharides produces polyelectrolyte complexes (PECs) through electrostatic complex coacervation (Figure 8) and hydrogen bonding. Coacervation can be achieved even without the use of chemical covalent crosslinking [85,107,108].

GM is a neutral, uncharged polysaccharide; hence, it has limited applications, but carboxymethylation gives it a negative charge that can interact electrostatically with positively charged polymers. CMGM showed optimum coacervation at pH 6.5 and a mass ratio of 1:1 with chitosan, and the results were positively related to DS. Higher DS provides denser tissue structure, smaller particle size, and greater elasticity [81].

Shi et al. prepared nanospheres by combining CMGM and 2-hydroxypropyl trimethyl ammonium chloride chitosan (HACC), a positively charged chitosan derivative, as a controlled-release carrier for ovalbumin vaccine (OVA). Molecular electrostatic attraction exists between the cationic quaternary amine group (–NH3^+^) of HACC and the anionic carboxyl (–COO-) of CMGM. Therefore, both macromolecular chains aggregate and coil, leading to the formation of insoluble CMGM/HACC composite nanospheres and the continuous release of the OVA vaccine for more than 24 h in vitro. The release rate decreased with higher concentrations of CMGM and HACC because increasing the amount of both the anions and cations made the internal structure stronger and inhibited OVA diffusion from the composite [23]. Wu et al. also examined the effect of using crosslinkers on drug release. Compared to un-crosslinked CMGM/CS nanogel, crosslinking showed 30% slower release in the first 2 h and 60% after 8 h. This is because the use of crosslinkers culminates in a more compact network of nanogels, which enhances the retention of curcumin. These results suggest that crosslinked CMGM/CS nanoforms can be developed into a sustainable delivery system [3].

Subsequently, CMGM has been extensively investigated as a carrier for colon targeted release systems. The application led to 1% drug release in simulated gastric fluid with pH 1.0, but up to 97% after 12 h in simulated colonic fluid containing the enzyme β-mannanase [13]. This is because CMGM is resistant to pepsin and trypsin but is hydrolyzed and degraded by β-mannanase. Therefore, it has great potential in the construction of a colon-targeted delivery system where drug release is influenced by enzymes [13,109]. 

## 9. Discussion 

Among numerous naturally occurring polysaccharides, GM has considerable potential, warranting further exploration. These polymers are abundantly available from natural sources. However, *Amorphophallus* contains considerable amounts of GM, up to 93% of dry weight, and is the main plant genus used for commercial production of GM [6]. This also indicates that individual species are the main factor that determines GM content and the quality of GM flour. Some other factors that affect the amount of glucomannan extracted are age of tuber, method of extraction, processing temperature, and solvent concentration for purification.

GM, which is abundantly available, shows potential as an excipient for solid dosing. GM has an orthorhombic unit cell pattern—its physical structure contains possible sliding planes, which may be responsible for increased crystal plasticity [10,110,111]. This is why it exhibits good compressibility, and it is not capped at higher compression pressures during decompression due to decreased relaxation stress. In addition, GM is free-flowing, as demonstrated by an angle of repose < 35° [99,112,113], making it suitable for use as a direct-compression excipient in tablet manufacturing.

Several studies have reported some of the physicochemical properties of GM. GM shows very high-water solubility, yet it forms a very viscous solution at pH 5–7 even at low concentrations. The reason for this is high water sorption by GM. However, water sorption is affected by the degree of acetylation of GM chains. Chemical and physical modifications have been made to obtain GM with desired properties.

Structurally, the simplest GM molecules consist of repeating monosaccharide units, such as glucose and/or mannose, with hydroxyl groups, which act as the main site of chemical modification. The hydroxyl groups can be changed through chemical reactions such as substitution, grafting, oxidation, and deacetylation, or by disrupting the original structure. In its application as an excipient in solid preparations, chemical modification of GM is usually aimed at modifying the viscosity, solubility, and tensile strength. However, the resulting change in properties depends on the DS. Higher DS results in lower viscosity because substitution of hydroxyl for carboxymethyl reduces interactions (intermolecular hydrogen bonds) between GMs, so it can be assumed that less water is entangled, which decreases GM viscosity. Then, the hydrogen bonds that have been disrupted cause a high decrease in crystallinity, thereby increasing the solubility of GM in water. On the other hand, low DS only increases hydrophobicity [6]. Regarding the tensile index, addition of 0.9% CMGM increased inter-fiber binding, which is attributed to increased tensile index and folding endurance [74].

As a direct-compression excipient, GM has the potential to be used as a filler–binder if it is co-processed both thermally and hydrothermally. Besides having good flowability and compressibility, native GM has high viscosity, so binding capacity is very strong, meaning it can be co-processed with another excipient that has good wetting properties and high porosity in order to increase water intake, which will aid tablet disintegration [105]. However, if GM is to be applied in a sustained-release tablet formulation, then GM can be combined with sodium alginate or HPMC in order to form intermolecular hydrogen bonds and physical entanglement between the two polymers, making an efficient membrane to inhibit water penetration, delaying drug release from the matrix [26,84].

## 10. Future Recommendations 

GM is a polysaccharide that has promise as an excipient for solid dosage forms, especially for direct compression due to its free-flowing nature and compressibility. Some applications of chemically or physically modified GM have been reported. Chemical modification is suggested to modify the solubility, viscosity, and mechanical properties of GM, while physical modification of GM is suggested to modify swelling ability and drug release from the matrix. Although chemical and physical modifications of GM have been studied, compared to other polysaccharides such as chitosan or alginate, the studies are not wide or deep enough. The mechanisms behind the effects of modifications on pharmaceutical characteristics, such as the relationship between structure and functionality/application of modified GM, are not clearly understood. Thus, the study of mechanisms of modified GM is necessary for its development as a potential pharmaceutical excipient. 

In recent years, a wide variety of innovative approaches to modify GM have been developed through non-contaminating physical modification methods (green methods) such as microwave heating, ultrasound-assisted and hydrothermal processes, and ball milling. In addition, exploration of other plants as sources of GM may also be conducted to create a wider range of functionalities, which also may expand applicability.

## 11. Conclusions

Recent studies on the effects of chemical modification of GM demonstrated several advantages, such as modified solubility and increased gel homogeneity, thermal stability, and mechanical strength of films or tablets through hydrogen bonding, polyelectrolyte complexation, and crosslinking. The characteristics of chemically modified products depend on the DS. Meanwhile, physically modifying GM by co-processing with other polymers enhances swelling ability and mechanical strength, and allows altered drug release from the matrix (e.g., through changes to hydrogen bonding and/or crosslink formation) without chemical changes. 

## Figures and Tables

**Figure 1 polymers-14-02550-f001:**
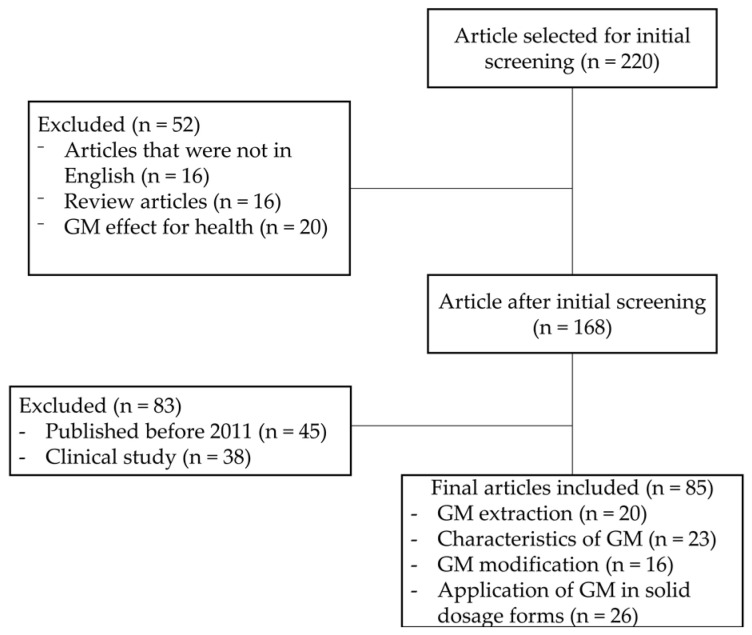
Flowchart of the inclusion and exclusion criteria for review articles.

**Figure 2 polymers-14-02550-f002:**
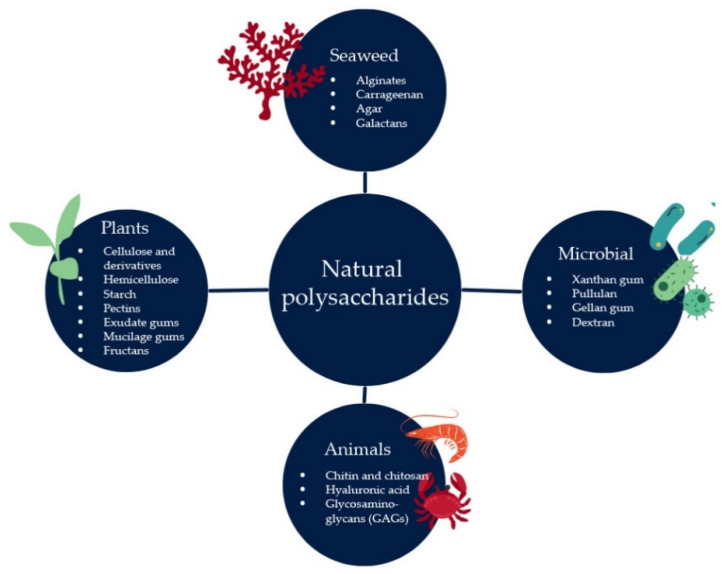
Four main sources of natural polysaccharides.

**Figure 3 polymers-14-02550-f003:**
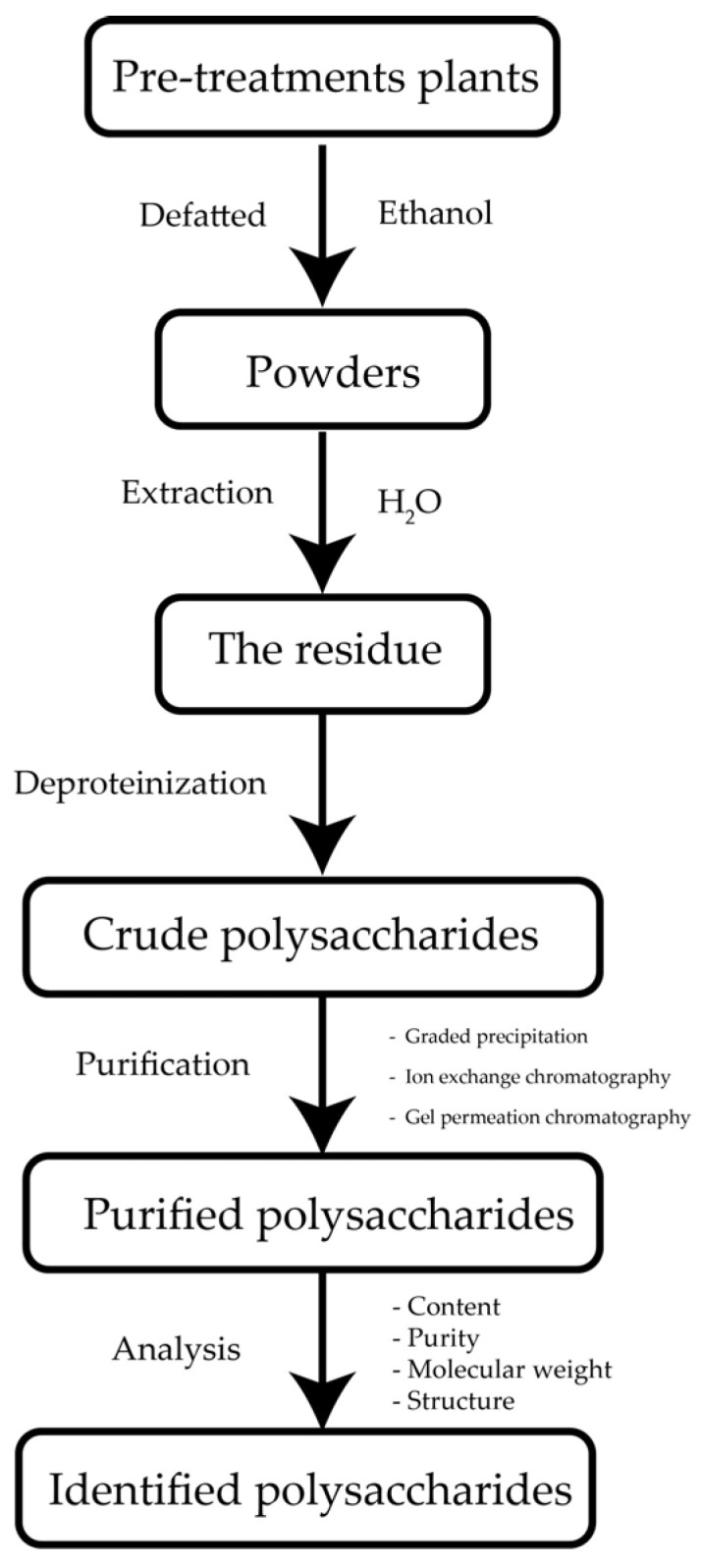
Isolation and purification of polysaccharides from plants.

**Figure 4 polymers-14-02550-f004:**
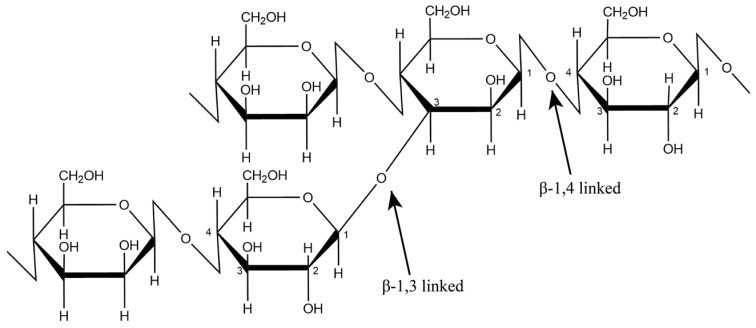
Structure of glucomannan.

**Figure 5 polymers-14-02550-f005:**
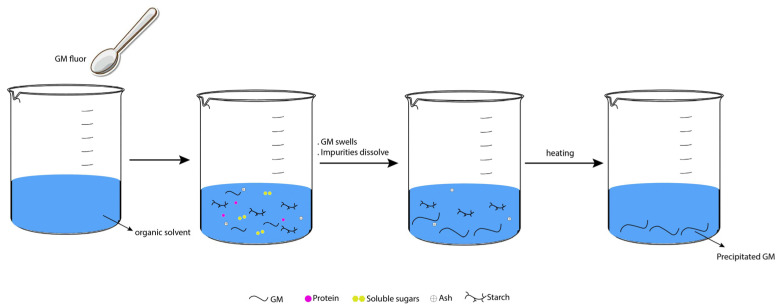
Separation of GM from impurities.

**Figure 6 polymers-14-02550-f006:**
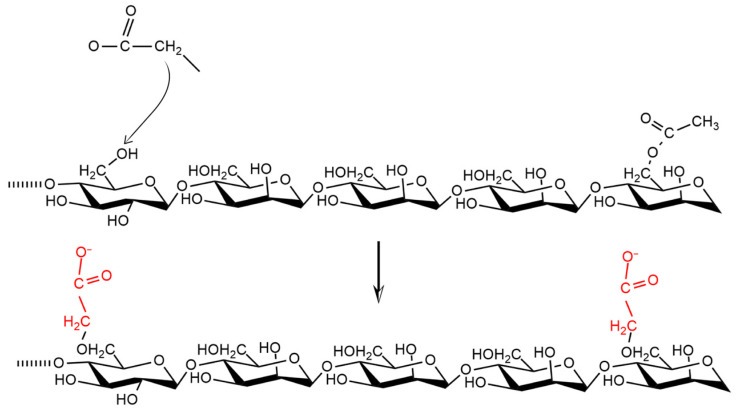
CMGM synthesis pathway.

**Figure 7 polymers-14-02550-f007:**
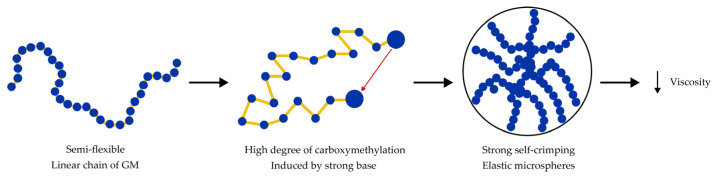
Effect of carboxymethylation on the structure of GM.

**Figure 8 polymers-14-02550-f008:**
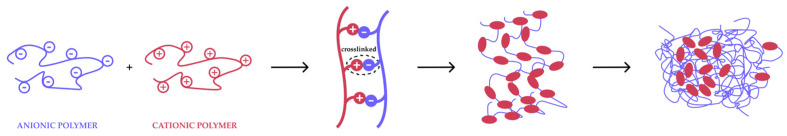
Formation of polyelectrolyte complexes (PECs) through electrostatic complex coacervation.

**Table 1 polymers-14-02550-t001:** Sources and extraction processes of GM from different crops.

Plant Sources	Part	Extraction Method	Principle	Extraction Solvent	Molecular Weight	% Yield	Ref
*Aloe barbadensis* M.	Leaves	Cold method (maceration for 24 h)	Maceration at room temperature with frequent agitation intended to soften and break the plant’s cell wall to release glucomannan	Ethanol precipitation	1.2 MDa	23.4%	[59]
*Amorphophallus muelleri* B.	Tubers	Cold method (maceration for 3 h)		Multilevel concentration of ethanol (40, 60, and 80%)	NA	62.2%	[55]
*Amorphophallus konjac*	Tubers	Cold method for 90 min		50% ethanol	9.5 × 10^5^ g/mol	91.4%	[60]
*Colocasia esculenta* L.	Tubers	Cold method with centrifugal rotational	Separation of starch and glucomannan is done by adding electrolyte salts such as NaCl to break the bond between starch and glucomannan Maceration at room temperature with frequent agitation intended to soften and break the plant’s cell wall to release the soluble glucomannan. Centrifugal rotational promotes the starch precipitate faster.	Isopropyl alcohol precipitation. Crude extract was extracted with water for 2 h	NA	4.08%	[61]
*Amorphophallus campanulatus* B.)	Tubers	Cold method with centrifugal rotational		Isopropyl alcohol precipitation. Crude extract was extracted with water for 2 h	NA	5.64%	[61]
*Salacca edulis* R.	Seeds	Hot water extraction (T = 95 °C for 2 h)	Glucomannan has greater solubility in hot water and is stable enough for minimum destruction with hot water extraction.	95% isopropyl alcohol solvent in a ratio (1:17)	2.057 × 10^4^ g/mol	40.19%	[62]
*Durio zeibethinus* M.	Seeds	Hot method (T = 95 °C for 2 h)		Isopropyl alcohol precipitation. Crude extract was washed with ethanol 95%	NA	39.60%	[63]
*Dioscorea esculenta*	Tubers	Hot method (T = 105 °C for 90 min)		Hot water extraction of the precipitate with isopropyl alcohol	1.865 × 10^4^ g/mol	53.09%	[64]
*Bletilla striata*	Tubers	Hot water extraction (T = 80 °C for 4 h)		95% ethanol precipitation. Crude extract was purified with DEAE-52 cellulose column	1.7 × 10^5^ Da	27.21%	[65]
*Amorphophallus oncophyllus*	Tubers	Hot water extraction (T = 55 °C for 1.5 h)		Purified with 95% ethanol	NA	93.84%	[6]
*Amorphophallus oncophyllus*	Tubers	Ultrasonic	Ultrasonic breaking of plant cell wall significantly improves glucomannan extraction efficiency	60% isopropanol	NA	59.36%	[66]
*Cibotium barometz*	Rhizomes	Alkali extraction	Glucomannan, a higher molecular weight polysaccharide, has greater solubility in dilute alkaline solutions than in hot water. Generally, extraction of the polysaccharides is first carried out in hot water and thereafter a dilute alkaline solution is employed for the extraction of residual polysaccharides.	Sodium hydroxide ([NaOH] 0.3 mol/L)	1445 Da	8.25%	[67]

**Table 2 polymers-14-02550-t002:** Co-processed GM with other excipients.

Combination of Excipients	Co-Processed	Application	Mechanism	Ref.
GM and HPMC K 100 LV	Microwave on level 5 (350 W) for 30 min	Matrix for gastro-retentive tablets forming a porous channel that allows the polymer mixture to absorb more water and expand, followed by prolonged drug release	Hydrogen bonds in single polymers have low energy, but the simultaneous formation of interlinked hydrogen bonds between polymer components provides significant interaction strength, resulting in a matrix that floats quickly and maintains the integrity of the polymer mixture under acidic conditions.	[97]
GM and lactose	Wet granulation	Filler–binder for direct compression of effervescent tablets	GM has a high viscosity and strong adhesive properties, thus providing good tablet binding effectiveness. GM has poor solubility in water, so it is combined with lactose as a water-soluble ingredient and to improve the poor flowability of lactose.	[98]
GM, sodium alginate (SA), and graphene oxide (GO)	Freeze dried	Microsphere-making polymers that enhance targeted delivery of drugs or nutrients to the colon	GM interacts with SA via hydrogen bonding and physical entanglement, and GO enhances these interactions in the microspheres. In addition, GO can greatly improve the loading efficiency of ciprofloxacin (CPFX) of microspheres, and achieve the sustained release effect of CPFX.	[26]
Oxidized GM, cassava starch, and sucrose esters	Dry heated	The OGM–CS combination exhibits low solubility and swellability, which makes it a possible excipient for the formulation of sustained-release drugs. However, the addition of SE significantly decreased porosity and swelling of the tablets, which inhibited immediate drug release.	Heating OGM and CS to high temperatures causes structural damage that limits the solubility and swelling ability of the polymer. The addition of SE with HLB 5 decreased porosity and slowed drug release because the more closed structure inhibited free movement of the drug out of the matrix. In addition, more hydroxyl groups in SE form hydrogen bonds, increasing intergranular bonding.	[84]
CMGM and 2-hydroxypropyl trimethyl ammonium chloride chitosan (HACC)	Complex coacervation and freeze dried	The coaservation complex formed can encapsulate and control the release of the molecular model for the vaccine, namely ovalbumin (OVA).	The anionic carboxyl group of CMGM and the cationic quaternary amine group of HACC cause intramolecular electrostatic attraction that causes the HACC and CMGM macromolecular chains to aggress and coil, forming the CMGM/HACC composite nanosphere.	[23]

## Data Availability

Not applicable.

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
