# Peer review of "Review on Modification of Glucomannan as an Excipient in Solid Dosage Forms"

_polymers, 2022, doi:10.3390/polym14132550_

Round 1

Reviewer 1 Report

polymers-1774844, Review on modification of glucomannan as an excipient in solid dosage forms

The manuscript of Nuur Aanisah and all is a resubmission of the paper polymers-1669649. The authors made some changes that are highlighted in red, but overall my comments from the previous review were in part ignored.

The authors added 3 or 4 references, but the total number is still low in my opinion. The authors should better present the information by adding a discussion section. The authors should add a more critical analysis of the data and its evolution. The review should not be just a collection of data, but a review of them. The authors need to present their opinion if there are gaps in the knowledge, the perspective of the use of glucomannan based polymers. What direction should the research take?

The authors need to discuss the existing similar review works in order to highlight the impact of this one. Does this manuscript add something new? Something relevant? Why is this work important as compared to the review works that already exist? The authors need to see this point in the manuscript.

The authors should better read the whole manuscript and present it better. It feels that various sections were written by various authors without reading the others sections. The manuscript has to be homogenous. It should be more clear and easier to read. 

Author Response

Response to Reviewer 1 Comments

Dear Editor and Reviewers of Polymers.

We thank the reviewer for reviewing the manuscript in detail and providing valuable suggestions for the improvement of our manuscript. We are pleased to submit the revision of the manuscript to Polymers. We have addressed each of the reviewers’ comments in detail below.

The manuscript of Nuur Aanisah and all is a resubmission of the paper polymers-1669649. The authors made some changes that are highlighted in red, but overall my comments from the previous review were in part ignored.

Point 1: The authors added 3 or 4 references, but the total number is still low in my opinion. The authors should better present the information by adding a discussion section. The authors should add a more critical analysis of the data and its evolution. The review should not be just a collection of data, but a review of them. The authors need to present their opinion if there are gaps in the knowledge, the perspective of the use of glucomannan based polymers. What direction should the research take?

Authors’ response: Thank you for your great comments. We have added the references used as the final articles included are 115 articles. Then, we have also added a discussion section which contains critical analysis from all the data collected. We reviewed glucomannan is because there are gaps in the knowledge, namely the abundant availability of natural polymers such as glucomannan which has the potential to be developed and enhanced by modification. However, the use of glucomannan as an excipient in solid preparations is still lacking when compared to other types of polysaccharides such as starch, cellulose, alginate and chitosan. These results provide insight for future research to apply both chemical and physical modifications of glucomannan as promising strategies for wider application in the pharmaceutical field. We sincerely hope that the revised manuscript has met the reviewer’s expectation.

After revision (line 497-546):

  1. Discussion

Among numerous types of naturally occurring polysaccharides, GMs are considered as a potentially polymer material to be explored. These polymers are abundantly available from the natural resources. However, Amorphophallus contain considerable amounts of GMs up to 93% of their dry weight and is the main plant genus used for commercial production of GMs [6]. These also indicated that the individual species is the main factor that determine the GM content and the quality of the GM flour. Some other factors that affect the amount of glucomannan during extraction are age of tuber, method of extraction, processing temperature and solvent concentration for purification.

GMs which abundantly available showed some potential to be developed as a solid dosage form excipient. It was observed that GMs have an orthorhombic unit cell pattern where its physical structure contains possible sliding planes and could be responsible for an increase in crystal plasticity [10, 112, 113]. This is the reason why this polymer exhibits good compression ability and are not capped at higher compression pressures during decompression due to the decreased relaxation stress. In addition, GMs exhibited free-flowing behavior as demonstrated by the angle of repose which is <35° [101, 114, 115]. This low angle of repose indicates good flowability, making it suitable for use as a direct compression excipient in tablet manufacturing.

Several studies have reported some of the physicochemical properties of GM. GM showed an a very high-water solubility, yet it forms a very viscous solution at pH 5–7 even at low concentration. The reason for this phenomenon is due to the high-water sorption of GM. However, the water sorption is affected by the degree of acetylation of GM chains. Modification efforts have been made to obtain GM with the desired properties, both chemically and physically modified.

Structurally, the simplest molecules of GMs consist of a monosaccharide repeating units such as glucose and mannose with hydroxyl groups, which is act as a main site of chemical modification. The hydroxyl group is changed through a chemical reaction such as substitution, grafting, oxidation, and deacetylation or by disrupting the original structure. In its application as an excipient in solid preparations, the chemical modification of GM is usually aimed at modifying the viscosity, solubility and tensile strength. However, the resulting change in properties will depend on the DS value. Higher DS results in lower viscosity because the substitution of hydroxyl groups in GM with carboxymethyl reduces interactions (intermolecular hydrogen bonds) between GMs, so it can be assumed that less water is entangled and decreases GM viscosity. Then, the hydrogen bonds that have been disrupted cause a high decrease in crystallinity thereby increasing the solubility of GM in water. On the other hand, if the DS obtained is low, it only contributes to an increase in hydrophobicity. Related to the tensile index, addition of 0.9% CMGM showed an increase in inter-fiber binding which is attributed to the increased tensile index and folding endurance [6].

In the application as a direct compression excipient, GM has the potential to be used as a filler-binder if it is co-processed both thermally and hydrothermally. Besides having good flowability and compressibility, native GM has a high viscosity so that the binding capacity is very strong so it can be co-processed with another excipient that has good wetting properties and high porosity because these attributes will increase the water intake, which will aid and increase the disintegration of the tablets [107]. However, if GM is to be applied as in a sustained release tablet formulation, then GM can be combined with sodium alginate or HPMC to then undergo the formation of intermolecular hydrogen bonds and physical entanglement between the two polymers which makes it act efficiently as a membrane for water treatment which delays drug release from the matrix [26, 84].

Authors’ response: The future research direction regarding glucomannan based polymers have been mentioned in the section 10. Future recommendation

After revision (line 547-564):

  1. Future recommendation

GM is a promising polysaccharide as an excipient for solid dosage forms especially for direct compression due to its free-flowing and compressibility behaviour. Some applications of modified GM have been reported either chemical or physical modification. Chemical modification is suggested to modify the solubility, viscosity and mechanical properties of GM, while physical modification of GM is suggested to modify swelling ability and drug release from matrix. Although chemical and physical modification of GM have been studied, however compared with other polysaccharides such as chitosan, alginate, the studies are not wide and deep enough. The mechanism of their modification effect on the pharmaceutical characteristics such as the relationship between structure and functionality/application of modified GM was not clearly understood. Thus, the mechanism study of modified GM is necessary to develop the GM as a potential pharmaceutical excipient.

In recent years, a wide variety of innovative approaches to modify GM have been developed through non-contaminating physical modification methods (green methods) such as microwave heating, ultrasound-assisted and hydrothermal processes or ball milling. In addition, exploration of other plants as sources of GM may also be conducted to create a wider range of functionalities which also may expand the applications.

Point 2: The authors need to discuss the existing similar review works in order to highlight the impact of this one. Does this manuscript add something new? Something relevant? Why is this work important as compared to the review works that already exist? The authors need to see this point in the manuscript.

Authors’ response: Thank you for your constructive comments. As far as we have read, there are several published manuscripts dealing with the chemical modification of glucomannan. However, in our review, we discuss not only chemical modification of GM but also physical modifications through co-processing and its relationship with application of GM as polymer excipients in solid dosage forms. Moreover, we also discuss about the factors of GM extraction from plants including harvest time, processing method, and solvent concentration. Owing to your great comment, we revised the manuscript as shown below.

After revision (line 383-496):

  1. GM application as an excipient for solid dosage forms

8.1. Direct compression excipient

Direct compression is the most preferred tablet manufacturing method because it is effective and efficient for industrial use. The excipient used in this process play an important role, especially in the formulation of low-dose active pharmaceutical ingredients (APIs), and is affected by the overall properties of the mixture. The excipients must meet requirements such as good flowability, compressibility, and compactibility.

GM can be a candidate for direct compression excipient because it has good flowability as demonstrated by the angle of repose which is <35° as well as the particle size of the CMGM produced namely 130 µm [24, 101].  Moreover, the results of co-processing with other excipients such as HPMC [99], lactose and starch [84] meet these criteria. This is because the co-process involves a combination of excipients thereby increasing the particle size which also affects compressibility. The smaller particles tend to have a larger air gap, which cannot be compressed during the process. The co-processed excipient with a larger particle size provides a more compact structure because less air is captured in the tablet and reduces the release of elastic energy [5, 87].

Based on X-ray results, deacetylated CMGM gave an orthorhombic unit cell pattern with a spatial plane a = 9.01 Å, b = 16.73 Å, c (fiber axis) = 10.40; and a possible space group of I222. Given that the orthorhombic crystals are not capped at higher compression pressures, this increased compressibility is due to high-level densification or decreased relaxation stress [10, 102].

8.2. Tablet disintegrants

A disintegrant is a substance or mixture of substances added to a drug formulation that helps disintegrate tablets into smaller particles, making them dissolve more quickly. Various natural polysaccharides such as modified starch, agar gum, and guar gum have been used as disintegrants [103].

Carboxymethylation of polysaccharides increase water absorption and cause faster disintegration time. As a comparison, sodium carboxymethyl cellulose (Na CMC) with a DS value of 1.0 tends to make CMC easily disaggregated in water making it more hydrophilic compared to water-insoluble cellulose [104, 105]. A similar property was also observed in the carboxymethylation of xanthan, where an increase in the degree of substitution indicates greater hydrophilicity and lower molecular weight [106].

GM is a hydrophilic natural polymer, but its solubility in water can be reduced by the formation of strong hydrogen bonds during the purification and drying processes. Several chemical modifications have been carried out to obtain derivatives with better solubility properties such as CMGM. The increase in solubility occurs due to the incorporation of water-soluble carboxylic groups. Changes in the water-binding properties are caused by the reduction and/or loss of crystal structure in the granules making them mostly amorphous and more hygroscopic [87]. Furthermore, the water absorption is influenced by DS, where carboxymethylation with a high value >0.5 helps increase absorption [9]. CMGM with a high-water absorption capacity causes rapid disintegration, hence, it has great potential as a tablet disintegrant. In contrast, low DS values ranging from 0.2-0.4 obtained with the sodium acetate catalyst reportedly reduced water absorption [87].

Ma et al. reported super-disintegration properties and rapid drug release by the combination of GM-superporous hydrogel composite (SPHC) on artificial gastric fluid media, where the drug release reached >90%, with GM accounting for 80% in the first 3 minutes. These results support the potential application of modified GM as a disintegrant for tablet dosage forms containing drugs with rapid disintegration [107].

8.3. Film-forming agent

One of the properties of GM solutions is their extremely high viscosity which needs to be modified either through chemical or physical modification. With a lower viscosity, this material is easier to apply when made into a film.

Xie et al. revealed that the combination of CMGM and cross-linked chitosan can be used in wound dressing due to the good swelling ability and moisture permeability which effectively protects the wound from excessive dehydration and accumulation of exudate. Moreover, the film produced had excellent thermal stability and biocompatibility, which can accelerate tissue regeneration [77].

Apart from wound dressing, the combination of CMGM with other polymers such as gelatin is used as packaging films with increased thermal stability. The single application showed continuous thermal degradation, while the initial temperature increased when it was combined with gelatin. Cross-linkage with gelatin through free amino and carboxyl groups also exhibited electrostatic interactions that improved the mechanical properties of the nanocomposite films [108].

8.4. Sustained release agent

The advantages of sustained-release tablets include increased plasma drug levels' stability and patient compliance, leading to optimum therapy [3, 23, 81, 84, 101]. One method to control drug release is through a polyelectrolyte complex reaction which improves and increases the gel strength. The mixture of two oppositely charged polysaccharides produces polyelectrolyte complexes (PECs) through the process of electrostatic complex coacervation (Fig. 8) and hydrogen bond formation. The coacervation of this complex can be achieved even without the use of chemical covalent cross-linking [85, 109, 110].

Figure 8. Formation of polyelectrolyte complexes (PECs) through the electrostatic complex coacervation process

GM is a neutral and uncharged polysaccharide; hence, it has limited applications, but modification by carboxymethylation changes the charge to a negative one which can interact electrostatically with other positively charged polymers. CMGM showed optimum coacervation conditions at pH 6.5 and a mass ratio of 1:1 with chitosan, but the results were positively related to their DS. Higher DS provides denser tissue structure, smaller particle size, and greater elasticity [81].

Shi et al., reported the preparation of nanospheres with the combination of CMGM and 2-hydroxypropyl trimethyl ammonium chloride chitosan (HACC), a positively charged chitosan derivative, as a carrier for ovalbumin vaccine (OVA) for controlled release. Molecular electrostatic attraction exists between the cationic quaternary amine group (–NH3+) of HACC and the anionic carboxyl (–COO-) of CMGM. Therefore, both macromolecular chains aggregate and coil, leading to the formation of insoluble CMGM/HACC composite nanospheres and the continuous release of the OVA vaccine model for more than 24 hours in-vitro. The release rate decreased with higher concentrations of CMGM and HACC because the increasing amount of both anions and cations made the internal structure stronger and inhibited OVA diffusion from the composite [23]. Wu et al., also examined the effect of using crosslinkers on drug release. Compared to the un-crosslinked CMGM/CS nanogel, the crosslinked type showed a slower release profile by only 30% in the first 2 hours and 60% after 8 hours. This is because the use of crosslinkers culminates in a more compact network of nanogels which enhances the retention capacity of the curcumin drug model. These results suggest that cross-linked CMGM/CS nanoforms can be developed into a sustainable delivery system [3].

Subsequently, CMGM has been extensively investigated as a carrier for colon targeted release systems. The application led to 1% drug release in simulated gastric fluid with pH 1.0, but liberated up to 97% after 12 hours in simulated colonic fluid containing the enzyme β-mannanase [78]. This is because CMGM is resistant to pepsin and trypsin but is hydrolyzed and degraded by β-mannanase. Therefore, it has great potential in the construction of a colon-targeted delivery system where drug release is influenced by enzymes [78, 111].

After revision (line 214-228):

Previous statistical studies which applied the Response Surface Methodology (RSM) proposed that the precipitation efficiency is correlated to several factors, namely harvest time [7], processing temperature [53, 54], and solvent concentration [53, 57]. First, the best harvest time to get GM from plants such as Amorphophallus muelleri tubers is when the condition is dormant, compared to before and after dormancy. For plants, dormancy is a period of arrested growth because GM as one of the energy sources for leaf growth is no longer used for metabolic processes and accumulates more in the tubers [58]. Second, the temperature correlates positively with the GM content according to Xu et al. which reported that the optimum temperature of 68°C using 40% ethanol increased the purity from 74.13% to 90.63% [53], while temperature of >78°C are not recommended because it is higher than the GM exothermic transition temperature and disrupt the molecular chain [54]. Third, the optimum concentration of ethanol solvent is 50%, this is because it is difficult to remove water-soluble impurities from the flour with a concentrated ethanol. Meanwhile, in diluted ethanol, more water is absorbed which leads to greater swelling, making it difficult to obtain the GM [53].

Point 4: The authors should better read the whole manuscript and present it better. It feels that various sections were written by various authors without reading the others sections. The manuscript has to be homogenous. It should be more clear and easier to read. 

Authors’ response: Thank you very much for pointing out this matter. We are aware of the confusion in the previous manuscript and so sorry for not making it clear. For example, the application of GM modifications, whether as a binder or disintegrant in tablets. Both can be applied. GM can be used as a binder because it has a high viscosity but must be combined with other excipients that has good wetting properties and high porosity through co-processing because these attributes will increase the water intake, which will aid and allow the tablet to disintegrate. We have modified the whole manuscript after reading the previous manuscript submitted to make our manuscript more clear and easier to read.

After revision (line 537-546):

In the application as a direct compression excipient, GM has the potential to be used as a filler-binder if it is co-processed both thermally and hydrothermally. Besides having good flowability and compressibility, native GM has a high viscosity so that the binding capacity is very strong so it can be co-processed with another excipient that has good wetting properties and high porosity because these attributes will increase the water intake, which will aid and increase the disintegration of the tablets [107]. However, if GM is to be applied as in a sustained release tablet formulation, then GM can be combined with sodium alginate or HPMC to then undergo the formation of intermolecular hydrogen bonds and physical entanglement between the two polymers which makes it act efficiently as a membrane for water treatment which delays drug release from the matrix [26, 84].

As the reviewer 1 suggest to undergo extensive English revisions, we have addressed the issue during revision. Our manuscript has been checked by native English-speaking/professional proof-readers; hence we include the certificate of it

Reviewer 2 Report

This review focuses on the modification of glucomannan (GM), as one polysaccharide, with chemical and physical modifications of the materials to enhance the water resistance and mechanical properties. This review is new and can provide unique perspectives in the modified GM studies. 

1. The introduction is not mature and too brief.

2. The section 2 methodology is oversimplified.
3. Can the author have a general summary of polysaccharide from different sources (e.g., extractions from plants or petrochemical synthesis) before section 3?

4. table 1 can be expanded to be more comprehensive - better include more classic papers. 

5. Figure 6 is unnecessary. 

6. section 5 chemical modification can be more detailed. 

7. The outlook should be more detailed. it is better for the conclusion and the outlook/future perspectives to be separated. 

Author Response

Response to Reviewer 2 Comments

Dear Editor and Reviewers of Polymers.

We thank the reviewer for reviewing the manuscript in detail and providing valuable suggestions for the improvement of our manuscript. We are pleased to submit the revision of the manuscript to Polymers.

This review focuses on the modification of glucomannan (GM), as one polysaccharide, with chemical and physical modifications of the materials to enhance the water resistance and mechanical properties. This review is new and can provide unique perspectives in the modified GM studies. 

We would like to thank to the reviewer for his/her kind appreciation and gracious comments. We are very pleased to hear that the reviewer finds our manuscpt interesting. We have addressed each of the reviewers’ comments in detail below.

  1. The introduction is not mature and too brief.

Authors’ response: Thank you for your careful review and helpful comments. Following to your suggestion, we added the introduction as shown below and we believe that the revised introduction has fully reflected a clear description of the article's scope, aims and structure.

After revision (line 42-73):

As a natural polymer, GM has properties that are superior to other polysaccharides when used as excipients for solid preparations, especially in tablet production. GM could be the excipient of choice for direct compression, the most efficient tablet manufacturing method, because it has desirable free-flowing and compressibility behavior [16–18]. GM is also reported as a largely used coating material and stabilizer in the pharmaceutical industry due to its gelling properties and particular rheological properties [11, 13, 19].

Native GM has several disadvantages for pharmaceutical application such as extremely high viscosity and low mechanical strength [20, 21]. In addition, GM’s high-water absorption index cause poor water resistance and limits some potential applications [14, 22]. However, these shortcomings of native GM could be modified through chemical or physical modification to enhance its structural and functional quality.

Chemical modification involves the substitution of functional groups in GM structures including esterification and etherification as well as elongation of the molecular chain through the formation of cross-links and encapsulation. The modifications alter the several characteristic of GM depending on the degree of substitution (DS) such as homogeneous film formation [11], increased tensile strength [15], improved thermal stability [15], and sustained release agent [23].

GM can be physically modified to improve functionality without undergoing chemical changes. The physical modification methods involve the mixing of native GM with other excipients under the influence of some physical factors such as milling [24], moisture [25], temperature [26], pressure [27], radiation [28, 29], etc. Physical modifications contribute to the variation in particle size, shape, surface properties, porosity, density and functional properties such as swelling capacity and gelation ability. These modifications directly influence disintegration and mechanical properties when used as an excipient in solid dosage forms.

Therefore, the objective of this review is to discuss several extraction methods of plant-derived GMs from different sources and. Meanwhile, the effect of chemical and physical modification of GM on its physicochemical characteristics are presented as well in order to explore its potentials, especially as a raw material for solid dosage excipients which are widely needed by the pharmaceutical industry.

  1. The section 2 methodology is oversimplified.

Authors’ response: Thank you for your great comments. Owing to your suggestion, we reorganized the section 2 methodology. We sincerely hope that the revised section has met the reviewer’s expectation.

After revision (line 72-78):

The review was based on studies identified in the Scopus and PubMed electronic databases using the keywords “glucomannan”, “modification”, “extraction” and “application” in the time range of 2011 to 2021. The specific keywords related to the application of glucomannan as the excipient of solid dosage forms were “chemical modification”, “physical modification”, “tablet” and “films”. Meanwhile, Reviews, studies written in non-English languages, original articles published before 2011, studies on unrelated topics such as glucomannan activity on health and its clinical studies were excluded. Abstracts were read and excluded if the reported article did not have any possible applications as excipients in solid dosage forms. Finally, 85 articles were fully read and included in the present paper. A flowchart of the article collection described in Figure 1.

Figure 1. Flowchart of the inclusion and exclusion criteria for review articles

  1. Can the author have a general summary of polysaccharide from different sources (e.g., extractions from plants or petrochemical synthesis) before section 3?

Authors’ response: Thank you for your comment. Owing to your suggestion, we added the the general summary of polysaccharide both from natural and synthesis for pharmaceutical application as shown below.

After revision (line 112-146):

  1. Polysaccharides from different sources

3.1. Natural polysaccharides

Natural polysaccharides can be derived from four main sources including algae, plants, animals, as well as microorganisms (Figure 2). The demand on polysaccharides that are from natural origin is tremendously growing because it provides several benefits including natural abundance, easy to isolate, and can be chemically modified in order to fit technological demand. Furthermore, these polymers may be hydrolyze through enzymatic reaction and considered to produce a noncarcinogenic degradable product [30, 31].

Figure 2. Four main sources of natural polysaccharides

Natural polysaccharides consist of many monosaccharide residues, which are joined by O-glycosidic bonds. When hydrolyzed, polysaccharides produce simple sugar units such as glucose, galactose, mannose, arabinose, xylose, uronic acid, etc. Among numerous sources of naturally occurring substances, the plant resources are considered as a potentially renewable resource group since they can provide a certain amount in supplying natural polymers of plant origin [32, 33]. Figure 3 shows some essential steps on the isolation and purification of polysaccharides from plants.

Figure 3. Isolation and purification method of polysaccharides from plants.

3.2. Petrochemical synthesis

Structurally, polysaccharides are composed of several monosaccharides joined together by O-glycosidic linkages. The O-glycosidic linkages is formed by the dehydration reaction of the hemiacetal hydroxyl group of one sugar (a glycosyl donor) with a hydroxyl group on the anomeric carbon of another sugar (a glycosyl acceptor). Due to the presence of multiple hydroxyl groups, one glycosyl acceptor residue can be connected to more than one glycosyl donor via different O-glycosidic linkages. Consequently, polysaccharides may be linear or branched, and branching may occur at different positions of sugar units in the polysaccharide backbone with different branching densities [34, 35]. Generally, three methods are used to synthesize polysaccharides: 1) stepwise glycosylation [36, 37]; 2) condensation polymerization [38]; and 3) ring opening polymerization [39, 40].

  1. table 1 can be expanded to be more comprehensive - better include more classic papers. 

Authors’ response: Thank you for your suggestions to our manuscript. We have modified Table 1 to make it easier to read and added the advantages and disadvantages of each method. In addition, we added some classic papers that discuss extraction of GM from other plants (Changes in page 6 line 232 which highlighted in red color).

Table 1. Sources and extraction processes of GM from different crops

Plant Sources

Part

Extraction Method

Principle

Extraction Solvent

Molecular Weight

% yield

Ref

Aloe barbadensis M.

Leaves

Cold method (maceration for 24 h)

Maceration at room temperature with frequent agitation intended to soften and break the plant’s cell wall to release glucomannan

Ethanol precipitation

1.2 MDa

23.4%

[59]

Amorphophallus muelleri B.

Tubers

Cold method (maceration for 3 h)

Multilevel concentration of ethanol (40, 60, dan 80%)

NA

62.2%.

[55]

Amorphophallus konjac

Tubers

Cold method for 90 min

50% ethanol

9.5 × 105 g/mol

91.4%

[60]

Colocasia esculenta L.

Tubers

Cold method with centrifugal rotational

Separation of starch and glucomannan is done by adding electrolyte salts such as NaCl to break the bond between starch and glucomannan with centrifugal rotational, the starch will precipitate faster Maceration at room temperature with frequent agitation intended to soften and break the plant’s cell wall to release the soluble glucomannan

Isopropyl alcohol precipitation. Crude extract was extracted with water for 2 hours

NA

4.08 %

[61]

Amorphophallus campanulatus B.)

Tubers

Cold method with centrifugal rotational

Isopropyl alcohol precipitation. Crude extract was extracted with water for 2 hours

NA

5.64 %

[61]

Salacca edulis R.

Seeds

Hot water extraction (T=95°C for 2 hours)

Glucomannan has a greater solubility in hot water and is stable enough so that with the hot water extraction method, glucomannan will get minimum destruction.

95% isopropyl alcohol solvent in a ratio (1:17)

2.057 x 104 g/mol

40.19%

[62]

Durio zeibethinus M.

Seeds

Hot method (T= 95°C for 2 hours)

Isopropyl alcohol precipitation. Crude extract was washed with ethanol 95%

NA

39.60%

[63]

Dioscorea esculenta

Tubers

Hot method (T= 105 °C for 90 minutes)

Hot water extraction the precipitated with isopropyl alcohol

1.865 x 104 g/mol

53.09%

[64]

Bletilla striata

Tubers

Hot water extraction (T=80°C for 4 hours)

95% ethanol precipitation. Crude extract was purified with DEAE-52 cellulose column

1.7 × 105 Da

27.21%

[65]

Amorphophallus oncophyllus

Tubers

Hot water extraction (T=55°C for 1.5 hours)

Purified with 95% ethanol

NA

93.84%

[6]

Amorphophallus oncophyllus

Tubers

Ultrasonic

Ultrasonic is used to break the plant cell wall which significantly improves the glucomannan extraction efficiency

60% isopropanol

NA

59.36%

[66]

Cibotium barometz

Rhizomes

Alkali extraction

Glucomannan, a higher molecular weight polysaccharide, has a greater solubility in dilute alkaline solutions than in hot water. Generally, the extraction of the polysaccharides is first carried out in hot water and thereafter a dilute alkaline solution is employed for the extraction of residual polysaccharides.

Sodium hydroxide ([NaOH] 0.3 mol/L)

1,445 Da

8.25%

[67]

  1. Figure 6 is unnecessary. 

Authors’ response: Thank you for your suggestions. We have deleted Figure 6 as the reviewer’s suggestion.

  1. section 5 chemical modification can be more detailed. 

Authors’ response: Thank you for your comment. We have added the discussion about chemical modification in section 5 chemical modifications following your suggestion. We sincerely hope that the revised section has met the reviewer’s expectation.

After revision (line 238-263):

  1. Chemical modification

Native GM formed very high viscosity solutions, where the intrinsic viscosity of 1% can reach 30,000 cps and so has potential use as a good film-forming agent [68]. However, a very viscous external gel layer on the surface of the particles immediately after dispersion prevents water penetration and drug dissolution and thus is limited to its use as a carrier for immediate drug release [69]. Then if made into film, it will turn into poor water resistance film due to its large number of free hydroxyl and carboxyl groups distributed along the backbone and exhibited high moisture absorption behavior. As a result, native GM has the weakness of poor water resistance and low mechanical strength [20, 21].

Several structural modifications for GM have been performed to enhance the structural and functional quality of these GM include oxidation [70–72] and etherification by addition of acetyl [41, 49, 59, 73] and carboxymethyl [2, 5, 13, 15, 74–80] moieties on hydroxyl groups of GM. Chemical modification with different degrees of substitution (DS) value give different physical and mechanical properties. The DS value itself was affected largely by the amount of sodium hydroxide and different dispersion medium. Higher degrees of substitution contribute to lower viscosity and particle size, denser network structure, and better tablet strength [81].

GM is an ideal candidate for being appropriately modified by chemical functionalization. Each of the glucose mannose units in has reactive hydroxyl groups which are the major sites for chemical modification. In addition, several studies discovered that chemically modified GMs can be used for the sustained release of drugs [2, 23, 43]. Among various modification methods such as acetylation [59, 73, 82], carboxymethylation and oxidation [70, 71, 83, 84], carboxymethylation is the most common and suitable for solid and film dosage forms [2, 3, 5, 74, 78].The effects of carboxymethylation on GM (CMGM) are described as follows:

  1. The outlook should be more detailed. it is better for the conclusion and the outlook/future perspectives to be separated. 

Authors’ response: Thank you for your suggestions. Owing to your suggestion, we have explored more the outlook to better present what direction should take in the future about GM. Also, the outlook and the conclusions have been separated.

After revision (line 547-576):

  1. Future recommendation

GM is a promising polysaccharide as an excipient for solid dosage forms especially for direct compression due to its free-flowing and compressibility behaviour. Some applications of modified GM have been reported either chemical or physical modification. Chemical modification is suggested to modify the solubility, viscosity and mechanical properties of GM, while physical modification of GM is suggested to modify swelling ability and drug release from matrix. Although chemical and physical modification of GM have been studied, however compared with other polysaccharides such as chitosan, alginate, the studies are not wide and deep enough. The mechanism of their modification effect on the pharmaceutical characteristics such as the relationship between structure and functionality/application of modified GM was not clearly understood. Thus, the mechanism study of modified GM is necessary to develop the GM as a potential pharmaceutical excipient.

In recent years, a wide variety of innovative approaches to modify GM have been developed through non-contaminating physical modification methods (green methods) such as microwave heating, ultrasound-assisted and hydrothermal processes or ball milling. In addition, exploration of other plants as sources of GM may also be conducted to create a wider range of functionalities which also may expand the applications.

  1. Conclusion

Recent studies on the effect of chemical modification on GM demonstrated several advantages such as modified the solubility, increased gel homogeneity, thermal stability and mechanical strength of films or tablets through the mechanism of hydrogen bond, polyelectrolyte complexation as well as cross-link formation. The characteristics of the chemically modified products depend on the DS value obtained. Meanwhile, physically modified GM by co-processing with other polymers enhance swelling ability and mechanical strength as well as modify the drug release from the matrix through the mechanism of hydrogen bond modification and cross-link formation without undergoing chemical changes.

Round 2

Reviewer 1 Report

The paper is improved and can be published.

Reviewer 2 Report

accept as is

This manuscript is a resubmission of an earlier submission. The following is a list of the peer review reports and author responses from that submission.

Round 1

Reviewer 1 Report

I warmly recommend a manuscript for publication in presented form. 

Reviewer 2 Report

This manuscript presented a review on modification of glucomannan as an excipient in solid dosage forms. However, the paper lacks perspective and critical discussion on the modification of glucomannan as an excipient in solid dosage forms. The content is not very relevant to the subject and not deep enough. In addition, there is a big problem that the content doesn’t corresponding with the references, like references [7], [8], [12], [13], [52], [55], [60] and so on. I suggest the authors check the references one by one to make sure it corresponding with the content. And the context of the article also need to adjust in order to express the meaning of the article. Therefore, I would like to recommend major revision, and some comments are shown as follows:

  1. The grammar is needed tocheck by a professional or native speaker. 
  2. Line 20, “CMGM”means what?
  3. Abstractneed to modify in order to express the meaning of the article clearly.
  4. L29, the authors write “coating materials”, however, the references was “gastric floating”, and “emulsifier [7]”, the references was “composite films”, “dan stabilizer [8]” was “cryogel”, which were not corresponding.
  5. L31, the authors wrote “In the last 20 years, the utilization of GM as a pharmaceutical excipient has become the lowest among all types ofpolysaccharides”, is there have references to support?
  6. For the whole introduction, I recommended that the author to reorganize the language, around the highlight and the point of innovation, in order to better express the meaning of the article.
  7. For “Structure and physicochemical properties”part, the content is not deep enough, mainly for popular science.
  8. Can the author explain the figure 3 in detail?
  9. Table 1, No 6 seems to the references was missing.
  10. L120-121, “In its original form, GM has several disadvantages such as an extremely high viscosity and low mechanical strength [12, 13]”, repeat with previous L40-41.
  11. L129, “carboxymethylation is the most common and suitable for application to solid and film dosage forms”, is there have references to support?
  12. L156, “ Meanwhile, pure GM has a contact angle of θ=48.1°”, however, in references[7], what the article mean was “Pure KGM film showed the lowest (48.1°) contact angles”.
  13. L168-171, the authors wrote “Matsumura, Nishioka, and Yoshikawa reported a dicarboxy-glucomannan derivative capable of increasing the solubility of GM in water and interacting with other positively-charged polymers”, however, I don’t find this conclusion in references [52].
  14. L131, the authors want to express “the effects of carboxymethylation on GM (CMGM)”, however, “2 Homogenous film formation”, pure KGM can also form homogenous film.
  15. L174, “Too high or too low viscosities produce a non-uniform film [55]”, I don’t find this conclusion in references [55], also.
  16. L187-189, “Deacetylation of GM through carboxymethylation changes the structure of GM from semi-flexible straight chains to elastic microspheres which decrease the inherent viscosity (Figure 5) [55]”, still don’t find this conclusion in references [55], also.
  17. L197-199, “The introduced COO− group is also able to efficiently bind more water which can be a plasticizer for the film to improve elongation”, still don’t find this conclusion in references [60], also.
  18. L225-226, the name of the table is?In addition, there is already table 1 in the L118.
  19. L321, “APIs”means what?
  20. L285-289, “Xie et al. revealed that the combination of CMGM and cross-linked chitosan can be used as a potential wound dressing since it exhibits good swelling ability and moisture permeability which can effectively protect the wound from excessive dehydration and accumulation of exudate. Moreover, the resulting film has excellent thermal stability and biocompatibility, which can accelerate tissue regeneration so that the in vivo results show wound healing[76]”, however, references [76] “the Carboxymethyl konjac glucomannan-crosslinked chitosan sponges for wound dressing” was “sponges”, not “film”.
  21. L316-325, “Chi et al reported the preparation of nanospheres with the combination of CMGM……[14]”, however, the references [14] was “Shi C, Zhu P, Chen N, Ye X, Wang Y, Xiao S……”.

Reviewer 3 Report

polymers-1669649: Review on modification of glucomannan as an excipient in solid dosage forms

The manuscript submitted for analysis is a review work that would be of interest for the pharmaceutical community, but to a lesser extent for the experts in the field of polymers.  The review presents the data from the last 10 years. It would be interesting for the readers to see if this new results advanced the field considerably or not. The authors should add a more critical analysis of the data and its evolution. The review should not be just a collection of data, but a review of them. Are there gaps in the knowledge? Does the use of glucomannan based polymers has a future? What direction should the research take?

There is a very low number of references. The authors eliminated the review work, but in my opinion they should discuss them in order to highlight the impact of this one. Does this manuscript add something new? Something relevant? Why is this work important as compared to the review works that already exist?

Row 71, the authors should state if they are taking about the natural polymer or chemical derivatives of it.

Row 103, “the condition is dormant, compared to before and after dormancy”. Please detail when does this happen.  What plant is discussed here?

Row 109, “increased the purity of GM from 74.13% to 90.63%” Compared with what method?

Table 1. row 2. Check the number format and editing. The same for rows 7 and 8.

Row “The incorporation of a carboxymethyl group reduces the linkage between the polymer chains”. Please detail in the paper if the “linkage” are hydrogen bonds or carbon oxygen bonds.

The section 140 to 171 is hard to follow. The authors should use a better plan to present the information.

The authors should better read the whole manuscript and present it better. It feels that various sections were written by various authors without reading the others sections. The manuscript has to be homogenous. It should be more clear and easier to read.  

The authors should provide also their point of view based on their experience in this matter.